# ODCS: On-Demand Hierarchical Consistent Synchronization Approach for the IoT

Safaa S. Saleh [1,*], Iman S. Alansari [2], Mounira Kezadri Hamiaz [3], Waleed Ead [4,5], Rana A. Tarabishi [2], Mohamed Farouk [6] and Hatem A. Khater [7]

1 Information Systems Department, Egyptian Institute of Alexandria Academy for Management and Accounting, Alexandria 21934, Egypt

2 Computer Science Department, College of Computer Science and Engineering, Taibah University, Medina 42353, Saudi Arabia; iansari@taibahu.edu.sa (I.S.A.)

3 Computer Science and Information Department, Applied College, Taibah University, Medina 42351, Saudi Arabia; mhamiaz@taibahu.edu.sa

4 Computer Science and Information Technology, Egypt-Japan University of Science and Technology (E-JUST), Alexandria 21934, Egypt; waleedead@bsu.edu.eg

5 Information System Department, Faculty of Computers and Artificial Intelligence, Beni-Suef University, Beni Suef 62514, Egypt

6 Department of Computer Science, Faculty of Computing and Information Technology, Arab Academy for Science, Technology, and Maritime Transport, Alexandria 21913, Egypt; mfaroukaast316@gmail.com

7 Mechatronics Engineering Department, Faculty of Engineering, Horus University Egypt, New Damietta 34518, Egypt; hkhater@horus.edu.eg

* Correspondence: safaa34@gmail.com

**Abstract:** An IoT data system is a time constraint in which some data needs to be serviced on or before its deadline. Distributed processing is one of the most latent sources in such systems and is considered a vital design concern. Many sources of delay in the IoT can affect the availability of data from different resources, which may cause missing data deadlines, resulting in a catastrophic effect. In fact, such systems are inherently distributed in nature and use distributed processing. The distributed processing permits different nodes to obtain the information from remote sites, which may take a long time to access the required data. Therefore, it is considered one of the most latent sources in such systems, which is considered a vital design concern. The typical recommended solution for this problem is to commit distributed transactions locally. Therefore, replication techniques are used to enhance the availability of data and consequently avoid the resulting latency. However, the use of local processing raises inconsistent periods. Therefore, this study proposes a new synchronization framework to minimize periods of temporal inconsistency. It permits several connected nodes to synchronize the shared data on demand concurrently without any need to use distributed synchronization, which consumes the system resource and raises its delay cost. The proposed framework aims to enhance the timely response of IoT real-time systems by minimizing the temporal inconsistency periods. The results indicate that the synchronization framework can be completed within a reasonable time period. They also depict improved consistency by minimizing the temporal inconsistency duration and increasing the chance of meeting critical time requirements.

**Keywords:** replication; temporal inconsistency; graph subnetting; on demand synchronize; IoT data

## 1. Introduction

The Internet of Things (IoT) is a promising technology that is only possible due to the technological innovations that have allowed for the reduction in components and extreme decreases in power requirements. It is used almost for enhancing data collection and automation using smart devices and technology. The IoT has become a key technology in many domains, such as healthcare, transportation, industry, and smart homes and cities.

The term IoT refers to a massive number of internet-connected "things" or nodes. In fact, the IoT is based on a large number of sensing nodes that act to collect different types of information from the surrounding environments via a variety of sensors [1]. The IoT uses many types of data communication, and real-time communication is the most well-known [2].

Time sensitivity is one of the most essential characteristics of the IoT. Some transactions in systems with complex timing constraints must be completed before the deadline for processed data [3]. Missing a deadline may have a fatal effect on the IoT system. The distributed nature of most IoT applications sometimes enforces the sharing of distributed data among several communicating nodes. In such systems, replication techniques are used to avoid delayed processing that can cause missing critical deadlines due to the delay in locating data from the remote nodes [4].

Using replication, when a datum is changed at a specific node (referred to by this work as the base node), many copies of this changed datum (replicas) need to be published to other consuming nodes. These consuming nodes are referred to by this work as replicated nodes where the new values are integrated after the propagation phase [5]. The most crucial design consideration for such systems is to make the required resources available in a reasonable time [6]. Many studies have been conducted to offer distributed real-time systems with strict criteria on the expected proper operation to improve the availability of IoT data. Their primary goal is to satisfy the temporal constraints of contemporary real-time IoT sensor networks [7].

Data synchronization in WSN is defined as a process of updating data on consuming nodes with the most recent changes made by the base node to maintain consistency (Yi et al., 2020). Concerning replication, data synchronization is responsible for updating local data with the propagated global values. The objective here is to ensure that all nodes use the same recent data. It has to be in real-time, and it is allowed in some cases to be near real-time. This process can be performed automatically with each change, which can increase the loads of the network traffic, increasing the latency [8]. Therefore, some works choose to link synchronization with demand [9]. In general, the most recent replicated data versions need to be synchronized by each consuming node. The IoT needs to accelerate data synchronization between the data producers (publisher or base nodes) and the consumers (subscriber or related nodes). The main challenge here is how to minimize the temporal inconsistency periods via different replicas of the collected information [10].

To avoid the delay resulting from the distributed committing, many solutions are directed to be based on the local committing. Viz, meeting time sensitivity comes at the expense of consistency requirements [11]. However, this technique can be accepted by such time-constraint systems to avoid the undesired latency, which can impact the critical time requirements [12]. In fact, such systems are allowed to temporarily relax the consistency requirement at the expense of availability. With the IoT, availability and reliability are more important than rapid universal consistency.

In principle, by using on-demand synchronization, the local versions of data items are synchronized with the most recent version only when they are needed (Raynal, 2018). It is utilized to minimize the need for remote synchronization that may impact the capability of IoT systems to meet their time requirements [13]. On-demand synchronization can be implemented locally using the most recent states stored on the local nodes. This scenario requires replicating the final state of the modified data items instead of transferring the updating transaction itself. These states are transferred via detached replication [10]. On-demand synchronization can also be implemented remotely (globally) by using the most recent states from the publisher node. However, searching the recent states is a time-consuming operation that can cause timing requirement violations [14].

The current study addresses the temporal inconsistency of time-sensitive IoT systems to enhance the consistency of such systems using an efficient synchronization approach. It aims to minimize the temporal inconsistency periods among the different copies of data on different nodes. The current study proposes a new on-demand hierarchical consistent

synchronization approach (ODCS) for IoT data systems. ODCS is based on the state transfer, not on the operation transfer with on-demand synchronization. The synchronization by ODCS is prioritized to be performed locally first. If the replicated versions in the local node are not sufficiently recent, the global synchronization starts with the specified base node.

To minimize the transmission delay, the ODCS needs to communicate with the publishing node via the lowest time–cost routing. Here, it represents the role of Network Time Protocol (NTP) with the graph subnetting mechanism. Although NTP is a clock synchronization protocol for distributed applications, ODCS (a data synchronization framework) benefits from its hierarchical method of evaluating the time difference between different network nodes. NTP, as an application layer protocol, is responsible for evaluating the delay between nodes of a TCP/IP network with the ability to support consistent timekeeping for data.

In fact, the introduced framework is motivated to support the IoT and similar systems to address the temporal inconsistency problem. The proposed on-demand framework includes two approaches for local and global synchronization. Using local on-demand synchronization is the first contribution of this work, where the most recent version is locally synchronized on demand. The IoT is supported in this fashion to avoid pointless update activities and the associated scheduling and problem-solving overhead. The second contribution introduces global on-demand synchronization to update the outdated items with the most consistent state from the responsible base node. The global on-demand synchronization method aims to reduce the time required to locate the global state using a virtual graph subnetting to produce a set of subgraphs to the same base node. Then, NTP helps in calculating the time cost of all subgraphs parallel acts to speed up the retrieving of the global state and consequently reduce the temporal inconsistency period. This can enhance the system performance, increase the chance of meeting the time necessities, and develop the consistency of the IoT.

The rest of this article is structured as follows: Section 2 discusses the earlier connected works. The proposed method for increasing the consistency of the IoT is presented in Section 3. Section 4 presents an experimental investigation to assess the suggested framework. Finally, Section 5 presents the conclusions of this study.

## 2. Related Work

Consistency is considered one of the most important topics when working with replicated data in distributed systems [15]. The main objective of the majority of academics in the IoT sector is to preserve the eventual consistency of time information by relaxing reliability at the cost of accessibility [16]. Many strategies with various ideas are proposed to equalize the inconsistency periods [17].

By surveying some research in on-demand synchronization, the article by [18] with the title On-Demand Depth-First Traversal comes first. The authors of this research utilize the concept of "On-demand" to decrease unnecessary operations in addition to the time-consuming processing. The proposed work has benefited from the idea of this work and the work by [11] to exclude unnecessary operations in light of timing requirements [19]. However, temporal consistency, which is the main concern of the proposed work, is not included in the work by [19].

Ref. [20] used an on-demand selective method, Freshness/Tardiness (FIT), to maintain consistency and scalability using state-transfer updating. To manage the trade-off between latency and consistency, running transactions are selectively scheduled such that pending changes are applied. For the most part, FIT relies on installing or skipping pending updates when using a state-transfer method. However, when using an operation transfer model, FIT effectively calculates the ideal number of updates that should be installed in order to maximize advantages. This work differs from ODCS as it focuses mainly on scalability, while the ODCS concerns mainly data availability. In addition, this work does not consider latency and energy requirements. The ODCS benefits from their idea to control the trade-off between latency and consistency.

To preserve the freshness of data in vehicle control systems, ref. [21] introduced an on-demand updates strategy for efficient CPU consumption and reduce unnecessary transactions. The introduced algorithm is based on a dynamically changed dependency graph to reduce the generated transactions. Their results depict a significantly decreased number of triggered transactions. This work is one of the most related works of ODCS as it is based on the concept of on-demand updates. This work differs from ODCS as it does not consider global consistency or meeting the power and timing requirements.

Another work that depends on on-demand computing is by [22]. By implementing the replication and de-replication mechanisms using the fewest possible messages needed to perform these operations, they suggest an on-demand Efficient Replication and De-replication technique (OERD). By discovering the assets and obtaining them to satisfy demands in a transparent manner, the OERD technique allows relocation options on file transfer to be made without any user involvement. Replicating the requested files is linked to the total number of requests which must reach the threshold value. Unlike ODCS, the time and power requirements are not considered in this work.

On the other hand, the following works are not based on on-demand computing, but they concern the timing requirement, which is an important aim of ODCS. These works also do not consider the power requirement. They aim to reduce the time duration of synchronization operations, such as the work by [23] that changed the predefined More–Less framework to maintain temporal consistency. In addition, the work by [24] proposes a new protocol to give more chances to meet the transaction deadlines. Their protocol depends on building Lists of available replicas dynamically during the propagation to tolerate the loss of update messages.

Additionally, the following studies are considered the closest to ODCS. For example, the work by [7] introduces a simple synchronous protocol that focuses on data availability. This work has an optimistic responsiveness in terms of system speed time-constraint requirements. Generally, their results show that it is not ideal for replicas due to the long offline period. It remains interesting for future work to produce more realistic synchronous models as well as practical solutions in them. Also, the work by [11] introduces an intelligent replication approach considering the time requirement. However, it does not consider the power requirement, so it cannot be suitable for WSN or the IoT. Unlike ODCS, this work also does not consider global consistency.

Also, for the synchronization studies, the work by [25] introduces a data-driven machine-learning model for the synchronization of demand and supply in retail supply chains. They try to forecast the demand and reduce satisfaction lead time. Another work by [26] proposes an effective framework for data synchronization and consistency management. It is based on an inverted index structure to synchronize data without central cache management. Additionally, the work by [27] introduces a lightweight synchronization algorithm for WSN with an aim to synchronize data measurements to minimize negligible overhead. Unlike the synchronization model of ODCS, these works do not consider the local consistency or the time constraint of data, although they act to minimize the synchronization time.

For the use of NTP, the work by [28] proposes an accurate time synchronization model for the IoT that is based on an enhanced version of NTP, which utilizes one resource-rich node on the mesh network and uses the border router to synchronize with a global time reference. The work by [29] is similar to ODSC as it uses the timestamp in the synchronization process. Their main aim is clock synchronization, which is different from the aim of ODCS of data synchronization. They employ the data timestamp in the estimation of the clock skew and offset, and ODCS employs the local data timestamp with NTP in the data synchronization process.

In summary, Table 1 lists a number of studies that can be considered close to ODCS. Each of them is similar to ODCS in several aspects and differs in certain aspects. A summary of the differences and commonalities between ODCS and related works is presented in

Table 1. For a better understanding of the following table, it is essential to consider the following terms:

- Local synchronization refers to synchronization that occurs inside each node using the most recent data version;
- Global synchronization refers to synchronization that occurs by requesting the most data version from its base node;
- The timing requirement refers to the latency constraint property that requires a response within a certain deadline or job management;
- On-demand computing refers to a delivery model that can perform a task when it is needed only.

**Table 1.** Summary of the closely related works.

| Work | Local Synchronization | Global Synchronization | Timing Requirements | On-Demand Computing |
|------|-----------------------|------------------------|---------------------|---------------------|
| [22] | ✗ | ✓ | ✗ | ✓ |
| [11] | ✓ | ✗ | ✓ | ✓ |
| [26] | ✗ | ✓ | ✗ | ✗ |
| [27] | ✗ | ✓ | ✗ | ✗ |
| [7]  | ✗ | ✓ | ✓ | ✗ |
| [25] | ✗ | ✓ | ✗ | ✗ |
| ODCS | ✓ | ✓ | ✓ | ✓ |

## 3. The Proposed Approach

### 3.1. Mathematic Formulation

Most IoT system applications use time-constrained data. A fresh-consistent datum ($d$) is defined according to Equation (1) as:

$$|TS(d) - TS_c| \leq t_0 \tag{1}$$

where $TS(d)$ is the timestamp when $d$ is used, $TS_c$ is the created timestamp (time of creation or updating), and $t_0$ is the accepted duration or (duration threshold). The value of $d$ at timestamp ($ts$) is expressed as $value_d^{ts}$. All data that are managed by this work have deadlines, and they can be used by sensor nodes, which are responsible for collecting the new values (referred to as the base node (B)), or other nodes that are referred to as replicated nodes (R). This work assumes that each datum can be modified by only one base node (B), and it can be used by many replicated nodes (R). Also, each base node can be responsible for collecting many data. Related nodes (or subscribers) of specified data are assumed to be registered by the corresponding base node in the subscriber list. As the (B) node modifies the datum, the new value must be propagated to all subscribers (Rs) of the modified datum. Figure 1 shows an example of the base nodes of different data items and related nodes of these data items. Note that Figure 1 notates (B) for the base node of any specified datum and uses (R) to represent the consumer node; therefore, it is notated as the related node for a specific datum.

In general, the nodes that make up this data's interconnections can be represented as a directed graph, G = (V, E), where V is the collection of nodes and E is the edges that signify the connections between those nodes. Figure 1 shows the whole data dependency graph G, which can be seen as a set of several subgraphs. Therefore, it can be split virtually into many smaller subgraphs to reduce the time consumed for traversing.

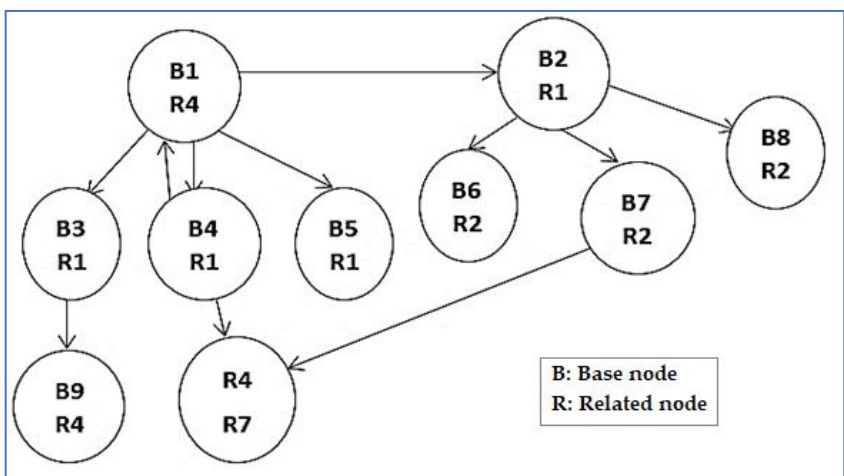

**Figure 1.** Network Graph.

Graph G can be dynamically changed by adding or removing nodes for many responses. Sometimes, sensors may die due to exhausting the residual power and need to be substituted with another powered sensor. Other times, extra sensors are needed to be added to the system based on certain situations according to the requirement of model change. These changes must be reflected in each corresponding base node. According to the current graph, each base node has a set of corresponding consumers (or replicated nodes) that need to receive replicated copies of each datum. Based on the proposed work, each base node (B) transfers the final states of its data to its specified consumers (R). The timing of ignition of updating or synchronization is determined by the timestamp of creation.

The process of the present work can be explained using the following example: A task that generates a new datum $d_i$ on the base node (as Figure 1: B1, B2, B3, B4, B7) has to transfer the new state of $d_i$ to its specified related (B3, B4, B5- for B1 and B6, B7, B8- for B2).

In fact, such time-constraint data systems need to deliver fresh data continuously to their consumers. The greatest problem for any work in this area is the trade-off between data integrity and achieving timing constraints in such a real-time IoT system. In order to increase consistency while reducing latency, this work suggests a tolerant, optimistic synchronization mechanism for the replicated real-time data of the IoT.

There are three kinds of transaction that handle information by each node according to the proposed approach: (1) the sensing transactions, which update the sensing data after each sensing operation; (2) the replicating transactions that act to publish the new values of the sensed data to their related nodes (destinations); and (3) the update transactions which are produced in the real-time system to keep the data fresh. These transactions are executed at each node based on their specified priority. The short-deadline data have higher priority than the data with longer deadlines.

There are two synchronization schemes for keeping the replicated data fresh: on-demand synchronization and immediate synchronization. The use of any of them depends on the timing requirements. For example, some applications need fresh data and cannot miss their deadline (as in hard real-time systems); here, immediate synchronization is more suitable for avoiding tardy transactions by missing the deadline. Other applications can be tolerant to the relaxed time requirement (such as soft real-time systems); this type can benefit from on-demand synchronization [8]. The proposed approach depends on two on-demand synchronization schemes: local on-demand synchronization and global on-demand synchronization. Using local on-demand synchronization, the synchronization with the most consistent state of data occurs inside each node using the most recent copy from the local replica repository. Global on-demand synchronization occurs when the most recent data version is out-of-date via requesting the most consistent state from the responsible base node.

When an application needs to use datum d, the real-time transaction here needs to obtain the freshest version of $d_i$ from the local repository or globally from the base node before going through the next step. At the reflection of this point, the proposed work needs to assess the consistency level of datum $d_i$ using Equation (2):

$$Con(d_i) = \frac{|t - TC(t_i)|}{t_0} \tag{2}$$

where *t* is the current time stamp (time of synchronization), *TC* is the creation timestamp of datum $d_i$, and $t_0$ is the accepted difference. Equation (3) can be used to decide whether a data item needs to be updated. The constraint for Equation (3) is as the following:

$$Con(d_i) must be < 1 \tag{3}$$

For global synchronization, ODCS needs to balance the impact of transfer time requirement (*TT*) and processing of synchronization time requirement (*TP*) as in Equation (4):

$$F(d_i) = w_1 * TT(d_i) + w_2 * TP(d_i) \tag{4}$$

where $w_1$ and $w_2$ are the weight of the transfer time and synchronization time and also satisfy $w_1 + w_2 = 1$. With the deformation and combination of (1) time transmission cost (*TT*) between the current node and the base node and (2) the processing time of synchronization (*TP*). $TT(d_i)$ is defined by Equations (5) and (6), and $TP(d_i)$ is defined by Equation (7):

$$TT(d_i) = \frac{TA(d_i) - TCC(d_i)}{t_0} \tag{5}$$

where $TA_{d_i}$ is the expected arrival timestamp of $d_i$ that is calculated by Equation (6).

$$TA_{d_i} = t + TCC_{d_i} \tag{6}$$

where $TCC_{d_i}$ is the communication time cost of $d_i$ between the current node and the base node:

$$TP(d_i) = \frac{|tp + TCC(d_i)|}{t_0} \tag{7}$$

where $TP(d_i)$ has to be $\leq 1$, *tp* is the processing duration of synchronization.

In a regular synchronization process, each node collects all out-of-date data items of the specific base node. Out-of-date data items have a difference in consistency level (Equation (3)) of more than one, i.e., $Con(d_i) \geq 1$.

As mentioned before, according to the proposed work, the traversing method considers that each data item ($d_i$) has only one base node, which is referred to as $B(d_i)$, and the base node can be responsible for many items. This will generate a large number of distributed synchronization transactions from the same base node. Therefore, the current approach planned to generate a list of required updates for current consumer node *R* from each base node *B*. In other words, if $d_i$, $d_j$ are two data that are required by the current node *R*, and the base node of $d_i$ ($B(d_i)$) is the same as $B(d_j)$, then ($d_i$, $d_j$) will be added into the same list of the same destination base node (*B*). This list is used to retrieve the most recent version of each out-of-date item from their base nodes.

If the current node has a direct edge with *B*, the consistency ($con_{cost R,B}$) cost between the base node (*B*) and the current replicated node (*R*) depends on the length of the out-of-date list (*list_Len*) that is expressed in Equation (8):

$$con_{cost R,B} = \sum_{i=1}^{list_{len}} Con(d_i) \tag{8}$$

If the current node (*R*) does not have a direct edge with (*B*), each node needs to revise all edges to (*B*), which is responsible for the required out-of-date list (*list*) using the depth-

first traverse. To decrease the workload of the running traversing transaction, the proposed approach may need to pre-generate a set of virtual subgraphs to each related (destination) node to form the indirect base node $\hat{G}(B)$ during the system preparation stage. This will help in the parallel evaluation of the cost of all subgraphs of the base node. This can enhance the global on-demand update requirements and reduce the temporal inconsistency periods. The constraint of choosing the best of all possible sub-graphs to base node $min\hat{G}(B)$ is expressed in Equation (9), where min-dd$_B$ is the minimum deadline for all required data items of the destination (*B*).

$$min\hat{G}(B) = \left| \sum_{i=1}^{list_{len}} O(d) \right| < (\min(dd_B)) \tag{9}$$

### 3.2. The Assumptions

The assumptions regarding the environment of the proposed framework are as follows:

1. The network is not fully connected, i.e., there is no point-to-point communication between the two nodes. In other words, global synchronization requires a long time and increases the inconsistency period;
2. All consumers of a given replica receive the new version automatically from the responsible sensor node and store it in its repository;
3. Each datum can be modified by only one base node and used by many replicated nodes. In addition, each base node can be responsible for collecting a large amount of data.

### 3.3. The Proposed Framework

According to ODCS, the two main factors that impact the temporal inconsistency periods of specific data between its base and consumer nodes are the time processing cost and time transmission cost. The time processing cost (TP) is the time that is consumed by executing the replicated transaction. The time transmission cost is the time that is consumed to transfer the replica between base and consumer nodes. ODCS aims at reducing the temporal inconsistency periods in two ways: the first way is to exclude the execution time of the updated replica (T3 from Figure 2). By eliminating the time required for transaction execution, transferring the final state of each datum rather than the entire update transaction would shorten the period an inconsistency lasts. The inconsistency periods (as shown in Figure 2) start from the end of the transaction committing on the producer (sender) to the end of the transaction committing on the consumer (destination).

A repository that is used to store the transferred replicas on each node is crucial to the consistent framework of ODCS. When synchronization is required, it provides the operating transactions with the most recent copy of the necessary datum and retains the received replicas. This repository is updated dynamically by receiving new versions at the expense of old ones.

The second means of ODCS to reduce the temporal inconsistency periods is to minimize the time transmission. It is based on minimizing graph traversing using the virtual graph subnetting to produce a set of subgraphs to the same base node. Evaluating each subgraph parallel will contribute to accelerating the selecting path and consequently reducing the temporal inconsistency period. This assessing technique is based on the hierarchical methodology of NTP to estimate the time cost of each resulting subgraph. ODCS allows data propagation without any global synchronization decision and acts to distribute the newest value of specified datums to their destination nodes.

Using the on-demand local synchronization, the running transaction compares the timestamp of the local version of the processed datum with the newest version in the replica repository to use the recent value. The local synchronization process begins at this point to decide whether to update the local version if the received version is more recent. Figure 3 summarizes this process of ODCS. If the difference between the resulting newest version and the current timestamp is more than the threshold, then the global synchronization starts to request the newest version from the responsible base node.

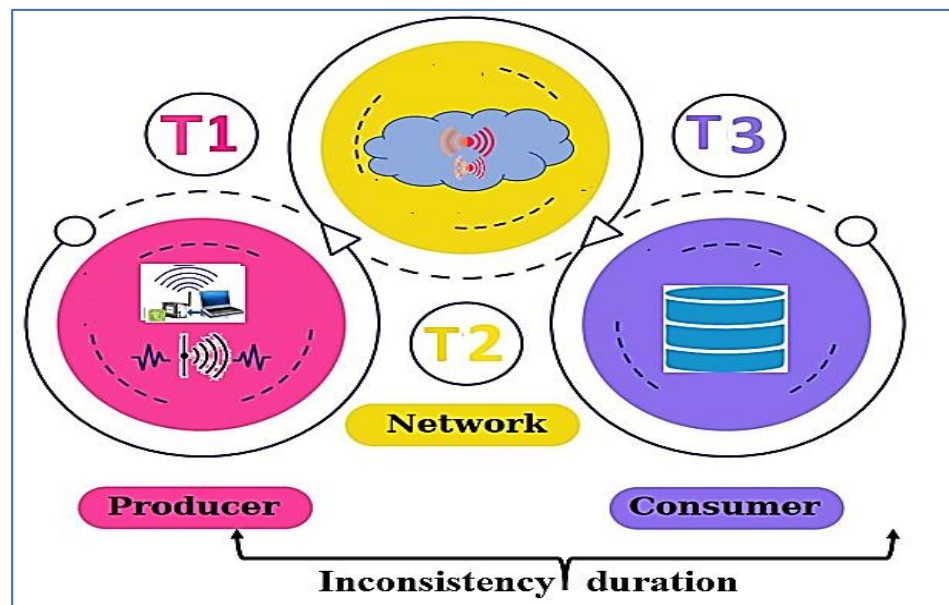

**Figure 2.** The temporal inconsistency periods. T1: producing time; T2: transmitting time; T3: executing time.

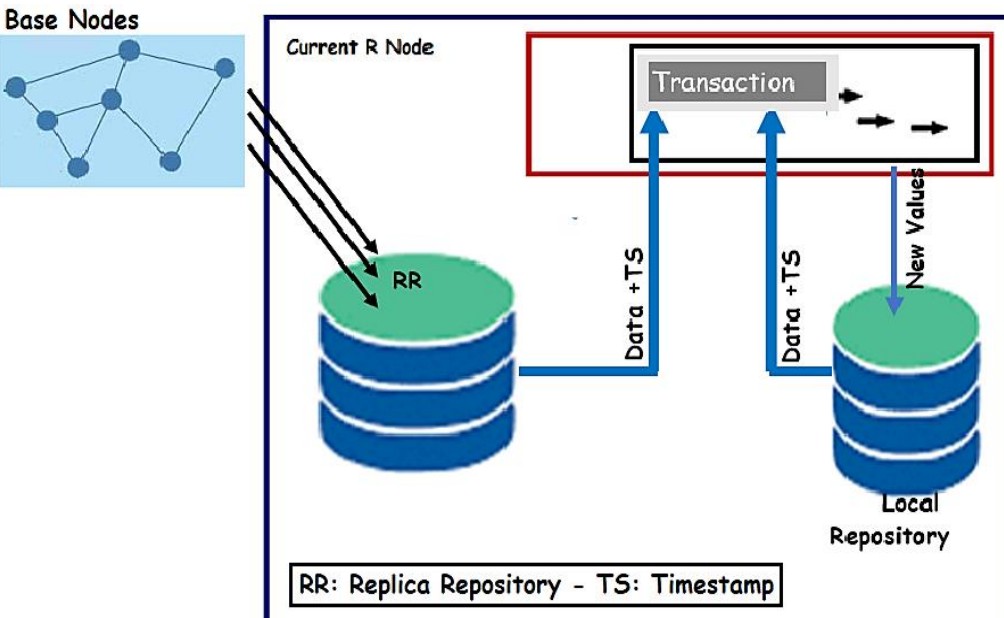

**Figure 3.** The Process of ODCS.

### 3.4. On-Demand Synchronization Algorithm

According to ODCS, the synchronization with the global data value is linked with any request of the individual datum locally. The synchronization procedure of the local value using the most recent version in the local replica repository is linked to the demand of the datum. To put it another way, whenever a datum is requested, the on-demand synchronization procedure is launched in order to obtain the most recent value of that particular datum and update the local value. Synchronization takes place only if it can suit the greatest degree of consistency, thanks to this optimistic process. Algorithm 1 offers the strategy for this approach.

| **Algorithm 1:** On-Demand Global Synchronization |
|---|
| Inputs: datum $d_i$ network graph schema GS, |
| 1.     Set counter I = 0 <br> 2.     Retrieve TS1 (di) from the local main repository. <br> 3.     If freshness (di) = true, then <br> 4.     Evaluate consistency cost//Equations (2)–(4) <br> 5.     Synchronize (di) from the local replica repository. <br> 6.     Else If freshness (TS1(di)) = false, then//Equation (1) <br> 7.     Use GS (sub-graphing)//Algorithm 2 <br> 8.     For each subgraph <br> 9.         For each node <br> 10.         Initiates a time-request exchange with the next node. <br> 11.         Calculate the link delay. <br> 12.         End For <br> 13.         Evaluate consistent cost//Equations (8) and (9) <br> 14.         Select subgraph with minimum cost.//Algorithm 2 <br> 15.         Update the graph schema. <br> 16.     End For <br> 17.     Retrieve TS(di) from the publisher node globally. <br> 18.     End if |
| Outputs: most recent value of $d_i$ |

Algorithm 2 is for traversing the network from the current replicating node (R) to the base node (B) using the breadth first. The main idea is to add several related nodes into a new graph that helps to select the minimum time cost of transmission transactions. For the target of keeping the previous relationship of nodes unchanged, the algorithm adds a new virtual smaller graph. The following pseudo-code is of Algorithm 2:

| **Algorithm 2**: Graph Subnetting |
|---|
| **Inputs:** graph G(d), Base Node (B), |
| 1.     Set Q []//queue for breadth-first traverse Sub_G[i]. <br> 2.     Set Nm//intermediate node <br> 3.     Set transCost[]//Transmission cost for Sub_G[] <br> 4.     For each possible starter of subgraph (Sub_G[i]) <br> 5.     Generate a new Sub_G[i] <br> 6.     Breadth-first traverse Sub_G[i] <br> 7.     Q[i].push(Nm) <br> 8.     If (Nm is not B), then <br> 9.     transCost[i] += delay transmission cost to Nm <br> 10.     Else continue <br> 11.     End if <br> 12.     End For |
| Outputs: set of Subgraphs from R node to B node |

### 3.5. Complexity Analysis

To evaluate the time complexity of ODCS, the two algorithms (Algorithms 1 and 2) presented in Sections 3.3 and 3.4 are considered. Algorithm 1 is responsible for retrieving the most recent version of data globally. Algorithm 2 is responsible for subnetting the routes from the base node to the related node. The performance of each algorithm is expressed by the time complexity. The time complexities of both Algorithms 1 and 2 are $O(n^2)$ and $O(n)$, where n is the total number of routes from the base node to its related node.

## 4. Experiments and Results

Using a customized JavaSIM simulator, a practical investigation to assess the suggested framework is conducted across the IoT system [30]. JavaSIM is a discrete event simulator that simulates sensor transactions and enables the generation of defined updates and synchronization transactions. For simplicity, a completely connected WSN system consisting of 10 nodes over different areas is used. The number of used data items is less than 100 items with time constraint features between (50–500 ms) and size equal to 64 bytes. The duration time of the sensor transaction is 30 ms. All nodes are used as base nodes, and all of them can be used as replicating nodes which generate synchronization transactions. Table 2 summarizes the set of parameters and the baseline settings for the simulation.

**Table 2.** Detailed Simulation Settings.

| Parameter | Default Value |
|---|---|
| Monitoring area | 100 m × 100 m |
| Number of nodes—N | 10/area |
| Number of data | <100 |
| Packet size | 6400 bits |
| Data deadline | 30 ms |

The results of ODCS are compared with some related on-demand and consistency approaches, such as BFT [7], IReIDe [11], and DOD [22], to assess the accuracy of the suggested research. In addition, the synchronization works by [25,27], and Lui and Lai (2018) are compared to evaluate the present work. Performance metrics are determined by [31] as useful indicators of better consistency are used to assess the results:

1. Transaction miss ratio (MR);
2. The throughput in terms of the number of committed transactions;
3. Total commitment delay;
4. Synchronization rate.

### 4.1. Transaction Miss Ratio (MR)

The number of missing transactions can be indicated by the number of arrived transactions to the total triggered transactions (update arrival rate).

$$\text{ArriRate} = \frac{AU_{\Delta t}}{U} \tag{10}$$

where $AU_{\Delta t}$ is the number of arrived transactions within the time threshold in time period $\Delta t$, and $U$ is the total number of updates. The arrival rate is evaluated in comparison with other related works, as depicted in Figure 4. This figure reveals that the miss ratio of ODCS is significantly better than its related works.

The outperformance of ODCS is due to the avoidance of global committing of transactions in addition to minimizing the time of transferring.

### 4.2. The Number of Committed Transactions

Throughput is another valuable metric for data synchronization and transmission. It is often used to compare against competitors for accurate evaluation. The observed throughput (the number of committed transactions/s) is depicted in Figure 5. This experiment examines the throughput for varying payloads in terms of number of replicas. Generally, the throughput of related works tends to be slightly less than ODCS. But as the number of replicas increases, the throughput of ODCS becomes closer to the works of [7,11]. This means that the throughput of all works becomes better with fewer number of replicas.

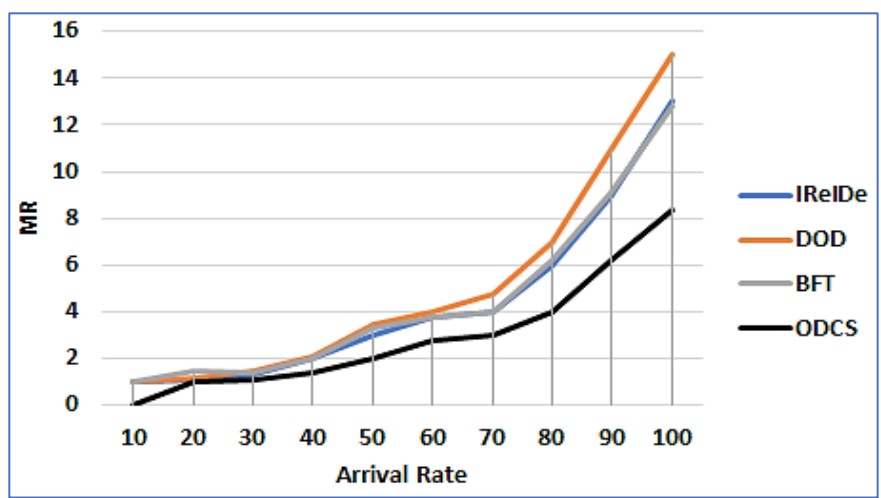

**Figure 4.** Transaction Miss Ratio (MR).

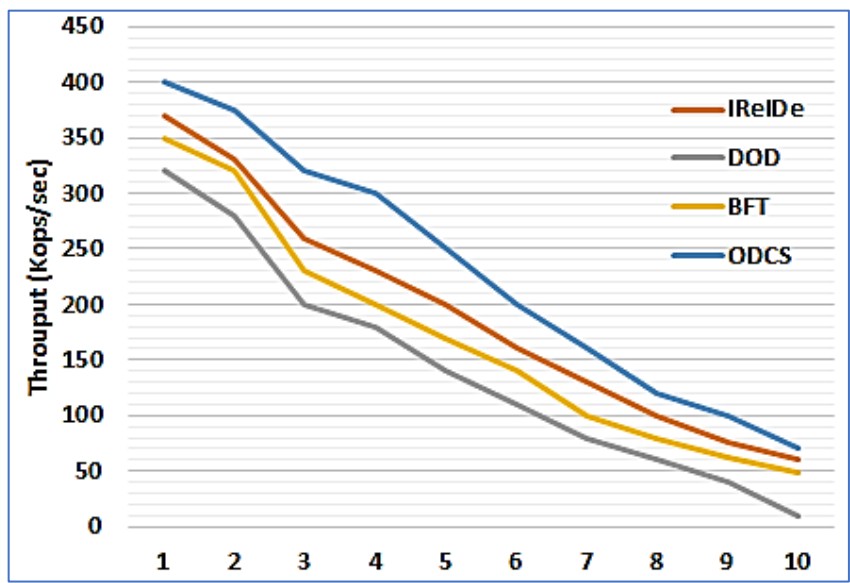

**Figure 5.** Throughput vs. number of replicas.

Unfortunately, the higher rate of replicas with lower throughput may point to the saturated state, which can lead to a communication bottleneck. As the replica load increases, the latency rises, causing the missing of deadlines; therefore, the number of committed transactions decreases.

### 4.3. Total Transaction Delay

The transaction delay is considered an essential metric to assess the eventual consistency. It indicates the transaction's commitment delay, which is defined as the time required to commit within its deadline [14]. To find the committed delay for successful transactions, this work customizes the updating transaction to obtain the beginning time, ending time, and data. Using the beginning and ending time, the duration of the transaction can be retrieved with different throughputs (10, 20, 30, 40, 50, 60, 70, 80) operation/sec at each node. Figure 6 depicts the results of the average duration with each throughput. The average maximum throughput is 0.88 ms, and the minimum validation duration is 0.75, which predicts the possibility of missing deadlines at the next trials. The figure shows the outperforming of ODCS, which reduces the penalty of missing deadlines compared to the related works, especially at an arrival rate of 80 updates/sec. This may be due to

the absence of impact of distributed committing in addition to the minimizing impact of transmission delay.

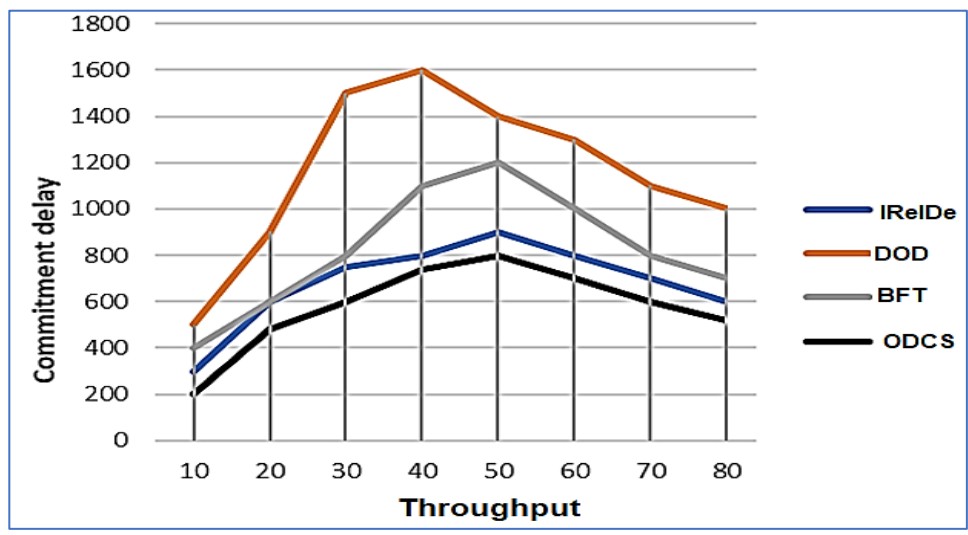

**Figure 6.** Commitment delay.

*4.4. Synchronization Rate*

To evaluate the performance of the introduced synchronization approach and its impact on data consistency, the synchronization metric is used. This metric is the successful update rate of an updated datum $(d_i)$ and is defined as

$$SucSynRate_{\Delta t}(d_i) = \frac{SU_{\Delta t}}{U} \tag{11}$$

where $SU_{\Delta t}$ is the number of successfully updated data within the time threshold in the time period $\Delta t$, and $U$ is the total number of updates involving data $(d_i)$. Figure 7 shows the outperforming of ODCS that considers the timing requirement of global synchronization.

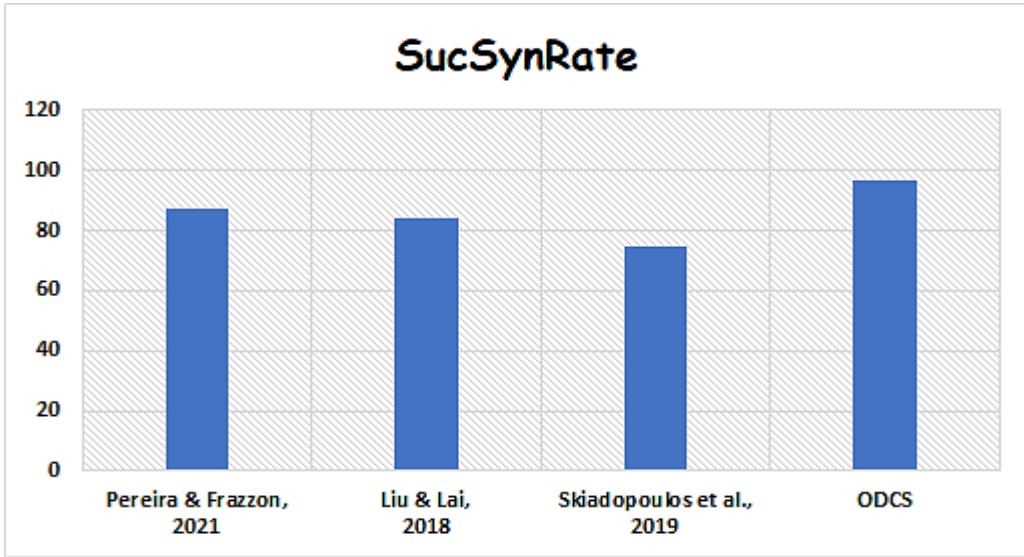

**Figure 7.** Successful Rate [25–27].

## 5. Conclusions

To maintain data freshness in the IoT system, an on-demand synchronization framework (ODCS) is introduced with local and global synchronization approaches. The synchronization operation of ODCS occurs on-demand locally when the datum is needed in any transaction. In this way, the IoT can avoid unnecessary update activities and the associated scheduling and problem-solving overhead. Global on-demand synchronization is used to update the outdated items with the most consistent value from the responsible base node. In this manner, the time of locating the global state is minimized using a virtual graph subnetting to produce a set of subgraphs to the same base node. The assessment method is based on the calculation of the delay of all subgraphs link by link parallelly. ODCS can successfully minimize the local inconsistency of replicated data by eliminating unnecessary update operations. The outcomes introduce the ability of ODCS to be completed within a reasonable time. They also depict improved consistency by minimizing the inconsistency periods. The experimental results show the ability of ODCS to reduce the number of missing transactions and the transaction delay by around 53% and 48%, respectively. In addition, the results also show the ability of ODCS to increase the throughput and synchronization rate by 33% and 10%, respectively.

For future work, it is planned to extend the proposed work by adding a new technique for continuous converging in the direction of a most consistent case, where conflicts are ignored as possible at the update level.

In fact, such systems suffer from another known problem, which is security. This problem is not considered by the current work, which mainly focuses on another problem in a different scope. Therefore, for future work, the authors plan to enhance the current work by adding a new component that complies, corrects, or prevents in some way the cyberattacks that could occur in these transactions.

**Author Contributions:** Conceptualization, W.E., M.F. and H.A.K.; Methodology, S.S.S., I.S.A., M.K.H., R.A.T. and H.A.K.; Software, I.S.A. and M.K.H.; Validation, M.K.H., W.E., R.A.T. and H.A.K.; Formal analysis, S.S.S., I.S.A., M.K.H., W.E., R.A.T., M.F. and H.A.K.; Investigation, M.K.H., W.E., M.F. and H.A.K.; Resources, S.S.S., I.S.A., R.A.T. and M.F.; Data curation, W.E.; Writing—original draft, S.S.S. and I.S.A.; Writing—review & editing, M.K.H., W.E., R.A.T. and M.F.; Visualization, S.S.S.; Supervision, S.S.S., M.F. and H.A.K.; Project administration, S.S.S. and H.A.K. All authors have read and agreed to the published version of the manuscript.

**Funding:** This research received no external funding.

**Data Availability Statement:** Data are contained within the article.

**Conflicts of Interest:** The authors declare no conflict of interest.

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
