# Peer review of "ODCS: On-Demand Hierarchical Consistent Synchronization Approach for the IoT"

_electronics, doi:10.3390/electronics12224708_

Round 1

Reviewer 1 Report

Comments and Suggestions for Authors

The paper is very well organized and all parts are well described. The authors give a good introduction supported by related work and compare the proposed approach with other similar approaches in the description and by comparing the results. The proposed approach is well described. The results are well presented and explained.

Comments on the Quality of English Language

The paper needs to be proofread. There are some minor errors, such as "by this way" that need attention.

The spelling of "on-demand", "On demand", "On Demand", etc. needs to be aligned throughout the paper.

Check the gray areas at the end of lines 142 and 166.

The sentences in lines 200 - 203 need to be reworded (instead of "for" use "is used" or "refers to".

Synchronize the spelling of di and di

Author Response

Comment

Response

·         The paper needs to be proofread. There are some minor errors, such as "by this way" that need attention.

Done

·         The spelling of "on-demand", "On demand", "On Demand", etc. needs to be aligned throughout the paper.

Done (On demand)

·         Check the gray areas at the end of lines 142 and 166.

Done (removed)

·         The sentences in lines 200 - 203 need to be reworded (instead of "for" use "is used" or "refers to".

Done

Lines 207-209

·         Synchronize the spelling of di and di

Done

Reviewer 2 Report

Comments and Suggestions for Authors

This paper shows that the ODCS could successfully minimize the local inconsistency of replicated data by eliminating the unnecessary updates operations.  But some results should be given in more details. Such as, the definitions of MR and arrive rate, also, the equations of them. 

Further, in Fig.5, the results show that the throughputs are more than 100, how to calculate the throughput values should be given.

Also, in Fig.6, the throughputs values are from 10 to 50, those are much smaller than the values from the Fig.5, the more explain should be given. 

Author Response

Comments of Reviewer #2:

Comment

Response

·         Some results should be given in more details. Such as, the definitions of MR and arrive rate, also, the equations of them. 

Done

lines 410-415

·         Further, in Fig.5, the results show that the throughputs are more than 100, how to calculate the throughput values should be given.

Response 1

·         Also, in Fig.6, the throughputs values are from 10 to 50, those are much smaller than the values from the Fig.5, the more explain should be given. 

Response 2

Response 1. Thank you for this important comment which leads to improved results. We have add how to  drive the throughput (the number of committed transactions/sec) in lines 421-423.

Response 2. Thanks for this valuable comment. The difference between figure 5 and figure 6 is that the throughput in figure 5 is used as a dependent variable (x-axis metric) but in figure 6, the throughput is used as an independent variable (y-axis) to evaluate the transaction delay.  

Reviewer 3 Report

Comments and Suggestions for Authors

The authors present a proposal to improve the response of IoT systems in real time, minimizing periods of temporal inconsistency.

The work is well written and presents scientific rigor.

The proposal is very relevant to the research area, demonstrating potential contribution to the area.

Therefore, the following comments are intended to contribute small suggestions for changes:

- the abstract has some parts that could be clearer, such as when the authors comment on distributed processing and its relationship with latency.

- in the introduction the authors state that IoT is a new advancement in wireless communication. Perhaps this expression is no longer so true given the maturity and time of use of this technology.

- throughout the text the expressions inconsistency period, inconsistency duration and temporal inconsistency are used. Perhaps it would be interesting to unify these expressions or make the difference between them clear (if there is one).

- the first word of the paragraph on line 167 has a larger font size than the rest of the text.

- better explain how the consumer nodes (the "Rs") in Figure 1 relate to the base nodes.

- sentence starting with a lowercase letter on line 260.

- in line 279 it seems that the equation for the TP term should be 7 and not 6. Likewise, the term TCC in line 281 does not appear in equation 7.

Reviewer 4 Report

Comments and Suggestions for Authors

The authors propose a framework to minimize synchronization inconsistency in IoT systems.

They begin by exposing a fairly complete introduction where concepts that will be used in the rest of the research are explained, to put the reader in context. They also provide the main contributions of the article.

Next, they take a detailed tour of the work related to the research provided. They continue with the proposal provided: mathematical formulation, assumptions in the proposed environment, the framework provided and the synchronization improvement algorithm, along with an analysis of the complexity.

They continue with a section where they show the results, facing the delays with the consistency achieved thanks to the algorithm provided.

They end with tight conclusions.

Despite being a complete work, it is proposed to review/update the following sections:

1- Many transactions occur on IoT platforms, and within them, inconsistency is one of the most relevant problems, which is discussed in the article. But there is another problem, which is security, which does not appear at any time in the article. Although it is not the main focus of the scope, the authors should make it clear in some way if the proposed algorithm complies, corrects or prevents in some way the cyberattacks that could occur in these transactions.

2.- To accompany the graphs, the authors may consider adding a complete table with exact data from the experiments.

3.- The conclusions should contain concrete data on improvement of the proposed algorithm. For example, when you say "ODCS can successfully minimize the local inconsistency of replicated data by eliminating the unnecessary updates operations", you could specify how much this improvement has been, in percentage or in any other magnitude, with respect to other studies.

4.- It would be interesting to propose some future work, with the proposed algorithm, or any other avenue of future research that the authors consider appropriate.

Round 2

Reviewer 2 Report

Comments and Suggestions for Authors

The notations in the figs, and parameters of the simulations for networking should be given in more details. 

Further, the successfully updated data should be closely related to the time threshold. therefore, different threshold would affect the performances.
